# Identification of the feature genes involved in cytokine release syndrome in COVID-19

**Bing Yang°, Meijun Pan°, Kai Feng, Xue Wu, Fang Yang‡\*, Peng Yang[ORCID]‡\***

The Second Affiliated Hospital, Guizhou University of Traditional Chinese Medicine, Guiyang, China

° These authors contributed equally to this work.
‡ FY and PY also contributed equally to this work.
\* 1874660968@qq.com (PY); 2267854716@qq.com (FY)

## Abstract

### Objective

Screening of feature genes involved in cytokine release syndrome (CRS) from the coronavirus disease 19 (COVID-19).

### Methods

The data sets related to COVID-19 were retrieved using Gene Expression Omnibus (GEO) database, the differentially expressed genes (DEGs) related to CRS were analyzed with R software and Venn diagram, and the biological processes and signaling pathways involved in DEGs were analyzed with GO and KEGG enrichment. Core genes were screened using Betweenness and MCC algorithms. GSE164805 and GSE171110 dataset were used to verify the expression level of core genes. Immunoinfiltration analysis was performed by ssGSEA algorithm in the GSVA package. The DrugBank database was used to analyze the feature genes for potential therapeutic drugs.

### Results

This study obtained 6950 DEGs, of which 971 corresponded with CRS disease genes (common genes). GO and KEGG enrichment showed that multiple biological processes and signaling pathways associated with common genes were closely related to the inflammatory response. Furthermore, the analysis revealed that transcription factors that regulate these common genes are also involved in inflammatory response. Betweenness and MCC algorithms were used for common gene screening, yielding seven key genes. GSE164805 and GSE171110 dataset validation revealed significant differences between the COVID-19 and normal controls in four core genes (feature genes), namely IL6R, TLR4, TLR2, and IFNG. The upregulated IL6R, TLR4, and TLR2 genes were mainly involved in the Toll-like receptor signaling pathway of the inflammatory pathway, while the downregulated IFNG genes primarily participated in the necroptosis and JAK-STAT signaling pathways. Moreover, immune infiltration analysis indicated that higher expression of these genes was associated with immune cell infiltration that mediates inflammatory response. In addition, potential therapeutic drugs for these four feature genes were identified via the DrugBank database.

**Data Availability Statement:** All relevant data are within the paper and its Supporting information files.

**Funding:** - Guizhou Provincial Key Technology R&D Program, ([2020]4Y178), Dr. Peng Yang -

The Fund of Guizhou Administration of Traditional Chinese Medicine, (QZYYXG-2021-12), Dr. Peng Yang - Guizhou Provincial Department of Education Research Fund, (2021-070), Dr. Peng Yang - The Fund of Guizhou Administration of Traditional Chinese Medicine, (QZYYXG-2020-5), Dr. Peng Yang.

**Competing interests:** The authors have declared that no competing interests exist.

## Conclusion

IL6R, TLR4, TLR2, and IFNG may be potential pathogenic genes and therapeutic targets for the CRS associated with COVID-19.

## Introduction

Coronavirus disease 19 (COVID-19) is an acute respiratory infectious disease caused by the novel coronavirus infection. The outbreak of the epidemic severely jeopardized the safety of the global population [1]. Fever, fatigue, and a dry cough represent the primary clinical manifestations. About half the patients developed dyspnea more than a week later, with rapid progression to acute respiratory distress syndrome, septic shock, refractory metabolic acidosis, and coagulation dysfunction in severe cases [2]. The damage to the body caused by viral infection is not limited to the virus itself [3]. A more serious effect is the systemic inflammatory reaction caused by the transitional immune system response to the virus, known as CRS. Several clinical studies have shown that CRS is an important cause of severe, critically illness, and even death due to COVID-19 [4, 5].

CRS refers to the excessive immune response caused by a dramatic increase in a large number of inflammatory cytokines [6]. Cytokines are soluble polypeptide proteins secreted by immune and non-immune cells, such as endothelial cells, epidermal cells, and fibroblasts, which play an intercellular regulatory role by binding corresponding receptors and controlling the inflammatory response, immune response, cell growth, differentiation, and maturation [7]. Cytokines can be divided into two categories according to the relationship between cytokines and inflammation. Class I represents pro-inflammatory cytokines, which can activate a variety of immune cells and promote inflammation. Class II includes an anti-inflammatory cytokine that neutralizes the effect of Class I cytokines, while the two interact to maintain immune system balance. In the case of severe lung infection in COVID-19 patients, cytokines can rapidly amplify the effect by binding to receptors, disrupting the homeostasis of the internal environment, leading to continuous lymphocyte and macrophage activation and amplification, and secreting a significant number of cytokines. This causes a series of pathological manifestations, such as endothelial dysfunction [8], systemic inflammatory response [9], coagulation dysfunction [10], and pulmonary fibrosis [11]. CRS is an important factor in the dangerous clinical manifestations of diseases, such as COVID-19, H5N1, and SARS, which can lead to death if not treated [12]. However, the CRS pathogenesis associated with COVID-19 remains unclear, while ideal intervention targets are lacking.

Disease occurrence and development is a complex biological process, which is often related to tens of thousands of genes and proteins. Therefore, it is extremely challenging to individually determine whether these genes are pathogenic using traditional methods. Bioinformatics is a new discipline involving the integration of life sciences, computer science, and informatics that uses massive biological data as the core and computers as tools for storage, retrieval, and analysis to reveal the related biological laws [13, 14]. In the big data era, a considerable amount of genetic disease data is stored in databases, the analysis of which helps to reveal the mechanisms underlying disease occurrence. This study conducts a secondary study on the COVID-19 data in the Gene Expression Omnibus (GEO) database by integrating bioinformatics. It also screens the CRS-associated disease genes from those in COVID-19 to provide an experimental basis for the pathogenesis and prevention of the CRS associated with COVID-19.

## Material and methods

### Data acquisition

The COVID-19-related dataset was retrieved from the Gene Expression Omnibus (GEO) database (https://www.ncbi.nlm.nih.gov/geo/). The training set information was obtained from the GSE164805 data of 10 COVID-19 patients and five healthy control patients (Healthy), while the validation set information was obtained from the GSE171110 data of 44 COVID-19 patients and 10 healthy control patients (Healthy). The dataset was downloaded using GEO query package (version 2.69.0) and the code used is listed in S1 and S2 in S1 Method. The CRS-related genes were searched in the GeneCards database (https://www.genecards.org/) using the keyword "Cytokine Release Syndrome".

### Analysis of the differential expression genes (DEGs)

The Principal Component Analysis (PCA) was conducted using the factoMineR (version 2.9), Factoextra (version 1.0.7) and ggplot2 (version 3.4.3), and the code used is listed in S3 in S1 Method. The DEGs were screened using the Limma package (version 3.57.7) and #dplyr package (version 1.1.2), and the code is listed in S4 in S1 Method, with $P<0.05$ and $|LogFC|>1$ as conditions [15, 16]. DEG volcano map was created using the org.Hs.eg.db package (version 3.17.0), dplyr package (version 1.1.2) and ggplot2 package (version 3.4.3). The code is listed in S5 in S1 Method. A DEG heatmap was created using the pheatmap package (version 1.0.12), and the code used is listed in S6 in S1 Method.

### Common genes and their functional enrichment

The DEGs of the COVID-19 and CRS disease genes were used to construct a Venn diagram to obtain common genes, which were subjected to GO and KEGG enrichment analysis using the ggplot2 (version3.4.3), pathview (version 1.41.0), clusterProfiler (version 4.10.0) and org.Hs.eg.db (version 3.17.0) package. The code used is listed in S7 in S1 Method, while $P<0.05$ was used for screening [17, 18]. Visualization analysis was performed on the top-ranked results to better understand the association between COVID-19 and CRS.

### Analysis of the transcription factors regulating common genes

Transcription factors are essential for transcription regulation. Most transcription factors recognize and bind specific DNA sequences to regulate spatiotemporal target gene expression. Common gene transcription factor analysis is vital for exploring the regulation of complex biological processes. The key transcription factors responsible for regulating target genes were downloaded from the TRRUST (https://www.grnpedia.org/trrust) and hTFtaret (hust.edu.cn) databases [19]. The DEG and key transcription factor interaction network was constructed according to the transcription factor-target gene relationship using the Cytoscape 3.8.2 software and subsequently visualized.

### Feature gene acquisition

The common genes were imported into STRING 11.5 (https://cn.string-db.org/). The species was set to "Homo sapiens" (human) to construct and visualize a Protein-Protein Interaction (PPI) network. The results were exported in TSV format and imported into the Cytoscape 3.8.2 software to construct a PPI network diagram. The network was then topologically analyzed, and core genes were screened using the Betweenness and Matthews Correlation Coefficient (MCC) algorithms [20–22].

### Verification of the feature genes

The feature gene expression information of COVID-19 and the Healthy groups were retrieved from the GSE171110 datasets, while the differences between the two groups were compared via ggplot2 package (version 3.4.3). The code used is listed in S8 in S1 Method, employing Student's t-test.

### The immune infiltration analysis of the feature genes

The ssGSEA algorithm in the GSVA package was used to quantify the immune cell proportion in the sample. The ssGSEA algorithm in the GSE164805 dataset was used for immune infiltration analysis, while the Wilcoxon test was employed to assess the immune infiltration expression differences between the COVID-19 and Healthy groups. The ccrrplot program and Spearman test were used to analyze the correlation between the feature genes and 28 infiltrating immune cells.

### Potential feature gene drug prediction

The DrugBank database (https://go.drugbank.com) is a comprehensive, free online resource. It contains detailed drug, drug-target, drug action, and drug interaction information of drugs and experimental drugs approved by the Food and Drug Administration (FDA), making it one of the most widely used reference drug resources in the world [23]. The DrugBank database was used to predict the core target regulatory drugs.

## Results

### Analysis of the COVID-19 and healthy DEGs

Comprehensive gene expression information was obtained from 10 COVID-19 patients and five Healthy controls in the GSE164805 dataset. The PCA of these data sets showed a PCA1 of 34.8% and a PCA2 of 20.5%, suggesting differences between the two data groups, which prompted difference analysis (Fig 1A). A total of 6950 DEGs were screened according to parameters including $P<0.05$ and $|LogFC|>1$, of which 2969 were upregulated, and 3981 were downregulated (Fig 1B and 1C).

### Common gene acquisition and functional enrichment

A total of 971 CRS-related genes were obtained from the GeneCards database, which was crossed with DEGs to obtain 34 common genes (Fig 2A). The KEGG analysis results showed that these common genes were mainly enriched in Cytokine–cytokine receptor interaction, necroptosis, and Toll-like receptor signaling pathways (Fig 2B), indicating that the signaling pathways involved in common genes were primarily related to inflammatory factor production. The GO enrichment results show that the common genes mainly involve receiver ligand activity and signaling receiver activator activity in the molecular function (MF) module, external side of plasma membrane in the cellular component (CC) module, and positive regulation of cycline production, positive regulation of defense response and positive regulation of interleukin-6 production in the BP module (Fig 2C). However, the remaining 164805 DEGs enriched in the top five signaling pathways showed significant differences compared to the 34 common genes (S1 Fig and S1 Table). This suggested that these common genes were involved in multiple biological inflammatory response processes during COVID-19 progression.

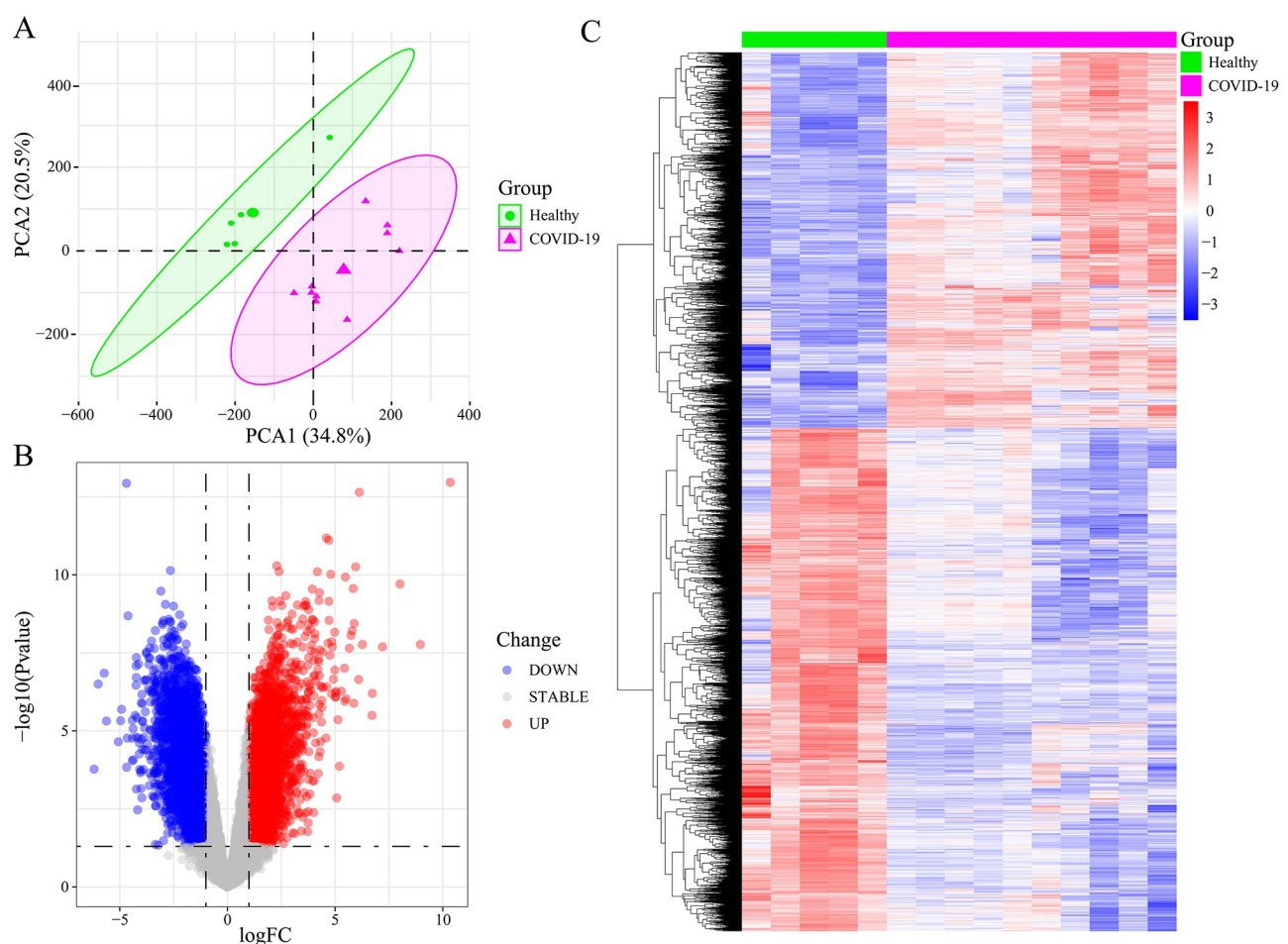

**Fig 1. Analysis of the COVID-19 DEGs.** (A) The Principal Component Analysis (PCA) of the COVID-19 (n = 10) and HC samples (n = 5). When PCA1 = 34.8% and PCA2 = 20.5%, COVID-19 and HC samples are clearly distinguished. (B) The 6950 DEGs volcano plots. The red dots indicate upregulated genes, the blue dots denote downregulated genes, and the gray dots represent non-DEGs, with FC≥1.0 and P-value<0.05. (C) The heatmaps showing the results of the clustering analysis based on 6950 DEGs.

## Analysis of the transcription factors for common gene regulation

To further understand the role of common genes in the progression of COVID-19 to CRS, their regulatory transcription factors were analyzed. A total of 26 transcription factors were obtained from the TRRUST database, while 354 were acquired from the hTFtaret database. After intersection, five transcription factors were obtained, namely Yin Yang 1 (YY1), Homo sapiens v-rel avian reticuloendotheliosis viral oncogene homolog A (RELA), early growth response 1 (EGR1), v-ets erythroblastosis virus E26 oncogene homolog 2 (ETS2), and interferon regulatory factor 1 (IRF1). The network diagram of the transcription factors regulating common genes was visualized using the Cytoscape 3.8.2 software (Fig 3).

## Key gene screening

In STRING11.5 analysis platform, the PPI network map was constructed using Cytoscape3.8.2 software, and the Betweenness and MCC algorithms were used to screen feature genes. (Fig 4A) shows the PPI network of common genes. The top 10 key genes obtained through the Betweenness algorithm are IL1 β, TNF, CD163, TLR4, INS, IFNG, CSF2, SERPINE1, TLR2,

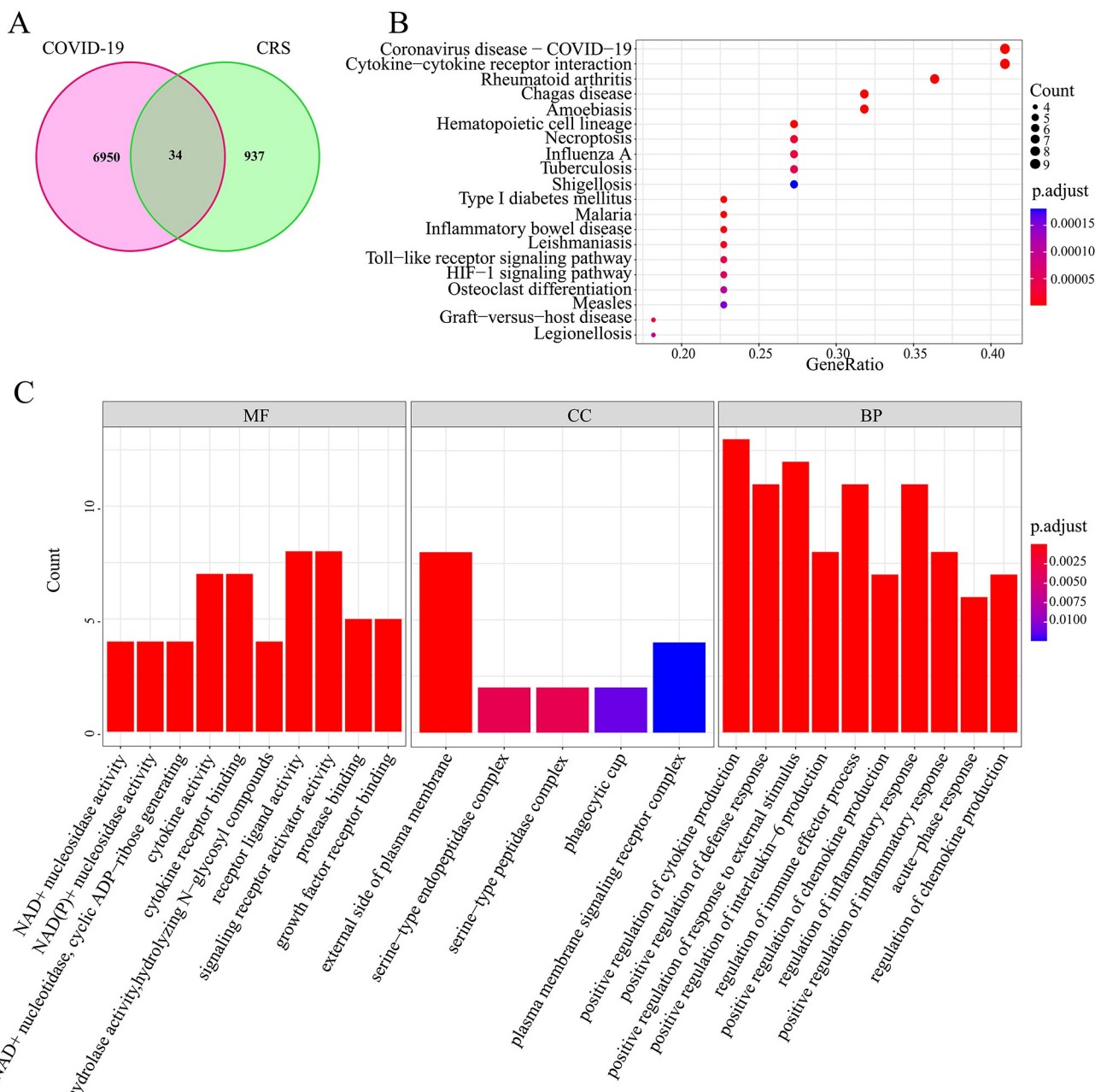

**Fig 2. The common genes and their functional analysis.** (A) The Venn diagram of the 6950 DEGs and 971 CRS genes. A total of 34 DEGs, namely common genes, were obtained. (B) The KEGG analysis of 34 common genes. (C) The GO analysis of 34 common genes.

and IL6R. The top 10 key genes acquired by MCC algorithm are IL1 β, TNF, IFNG, CSF2, TLR4, TLR2, IFNB1, CD28, IL1R1, and IL6R. A total of 7 feature genes including IL1 β, IL6R, TNF, IFNG, CSF2, TLR4, and TLR2 were identified after crossing the two algorithms (Fig 4B).

## Verification of the key gene expression

The key feature genes were further screened by searching the database. In the GSE164805 dataset, seven genes significantly differed from the Healthy group. The CSF2, IFNG, and IL1B

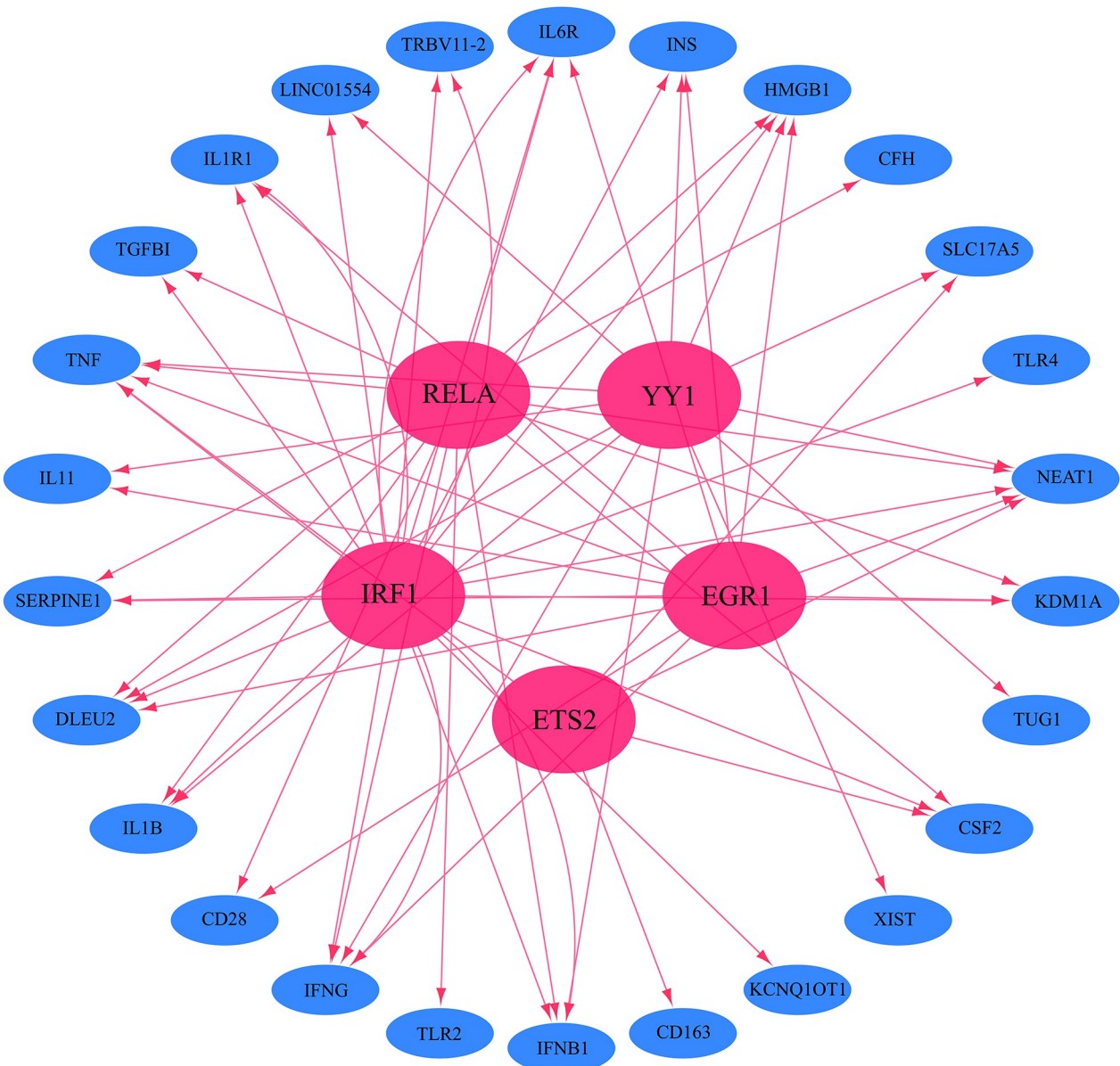

**Fig 3. The network map of the common genes regulated by transcription factors.** The 34 common genes are mainly regulated by 5 transcription factors RELA, YY1, IRF1, ETS2, and EGR1. Red represents the transcription factors, while blue denotes the common genes.

expression levels were considerably lower in the COVID-19 than in the Healthy group, while those of IL6R, TLR2, TLR4, and TNF were higher (Fig 5A). No significant differences were evident between the TNF and IL1B feature gene expression levels of the two groups in the GSE171110 dataset (Fig 5B), while CSF2 was not expressed in all COVID-19 samples. Therefore, the TNF, IL1B, and CSF2 genes were excluded. The remaining IL6R, IFNG, TLR4, and TLR2 genes (feature gene) were the most likely COVID-19 pathogenic genes that progressed into CRS.

## Analysis of the feature gene PPI network and KEGG enrichment

The PPI network analysis was performed using the GeneMANIA database, while KEGG pathway analysis was conducted on the feature genes and their associated genes. (Fig 6A) shows

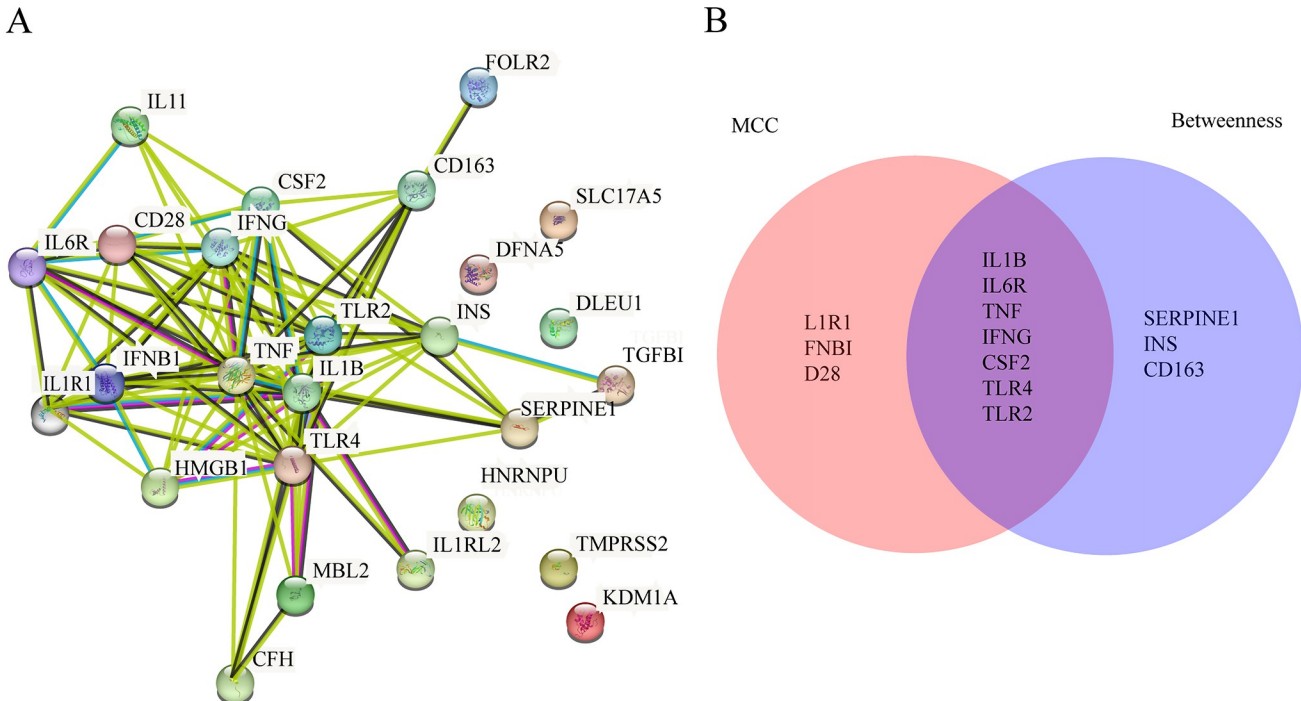

**Fig 4. Key gene screening.** (A) PPI network of common gene. There are 26 nodes with 92 edges, and the average degree value is 7.08. The size of the nodes in the figure and the depth of their colors are proportional to their degree value. (B)Venn diagram shows the intersection of the results of two algorithms.

the PPI network diagram of the upregulated IL6R, TLR4, and TLR2 genes, and (Fig 6B) shows that of the downregulated IFNG gene. The Toll-like receptor pathway was mainly involved in the upregulated and associated genes (Fig 6C), while the necroptosis and JAK-STAT signaling pathways were primarily involved in the downregulated IFNG and related genes (Fig 6D). These signaling pathways may be mainly related to the inflammatory response.

## Immune infiltration analysis

The ssGSEA algorithm was used to analyze 27 types of immune cell infiltration to determine their impact on the feature genes. The results showed significant differences in the GSE164805 dataset between the COVID-19 and Healthy groups regarding 13 types of cells, including effector memory CD8$^+$T cells, eosinophil, macrophages, neutrophils, plasmacytoid derivative cells, T follicular helper cells, and Type 17 T helper cells, P<0.05 (Fig 7A). The role of four feature genes in immune cell infiltration is shown in (Fig 7B). The higher TLR4 expression was positively correlated with eosinophil, macrophages, neutrophils, T follicular helper cells, and Type 17 T helper cells, and negatively with activated CD8$^+$T cells, central memory CD4$^+$T cells, and effector memory CD8$^+$T cells. The higher IFNG expression was positively associated with activated CD8 T cells, activated dendritic cells, CD56bright natural killer cells, CD56dim natural killer cells, central memory CD4$^+$T cells, and factor memory CD8$^+$T cells, and negatively with eosinophil, macrophages, neutrophils, plasmacytoid dendritic cells, and Type 17 T helper cells. TLR2 was mainly positively correlated with eosinophil, immature dendritic cells, macrophages, neutrophils, plasmacytoid dendritic cells, and Type 17 T helper cells, and negatively with activated CD8$^+$T cells, CD56dim natural killer cells, and factor memory CD8$^+$T cells. The higher IL6R expression was positively associated with eosinophil, immature dendritic cells,

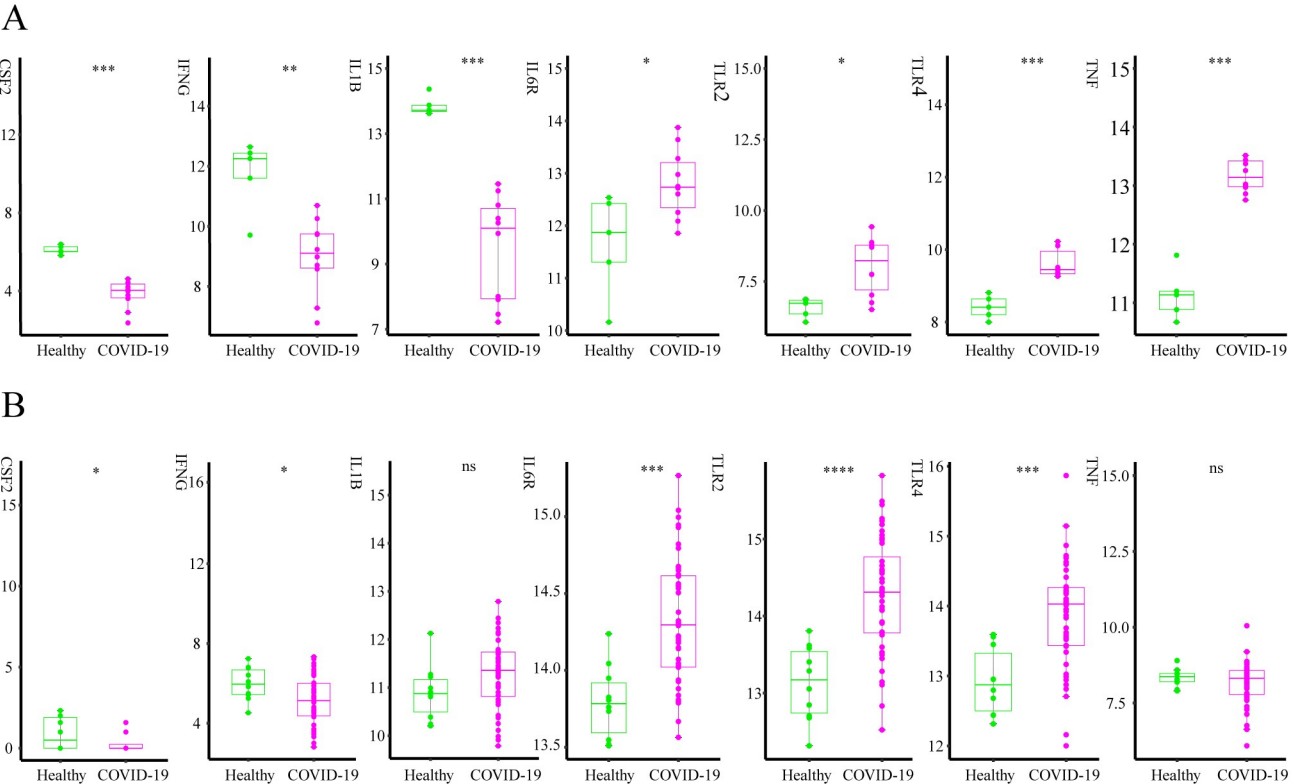

**Fig 5. A comparison and validation between the key genes of the COVID-19 and healthy groups.** (A) A histogram of the feature genes in the GSE164805 dataset. * indicates p<0.05, ** indicates p<0.01, and *** indicates p<0.001. (B) A histogram of the feature genes in the GSE171110 dataset. *indicates p<0.05, ***indicates p<0.001,and ****indicates p<0.0001.

macrophages, neutrophils, and T follicular helper cells and negatively with activated CD8[+]T cells. These results suggested that IL6R, TLR4, TLR2, and IFNG regulated multiple immune cell infiltration processes during COVID-19 progression.

### Prediction of potential therapeutic feature gene drugs

The DrugBank database was used to predict potential feature gene drugs for further analysis. The results showed that foreskin keratinocytes acted against IFNG, Tocilizumab inhibited IL6R, Eriodan restricted TLR4, and Adapalene impeded TLR4 Table 1.

### Discussion

This study analyzed a training set (GSE164805) to obtain a total of 34 CRS-related DEGs and four regulatory transcription factors from COVID-19 patients. After the validation set (GSE171110) analysis, only four genes, namely IL-6R, TLR4, TLR2, and IFNG, showed expression changes similar to those of the training set in COVID-19. Further analysis revealed that these four genes were key targets for inhibiting inflammatory factor production. These findings help reveal the CRS pathogenesis associated with COVID-19.

CRS is one of the main causes of deterioration and mortality in COVID-19 patients [24]. Although this study screened 34 genes associated with CRS (common genes) from COVID-19 patients, those primarily responsible for CRS progression and the role of different cytokines remain unclear. Enrichment analysis showed that these 34 genes were involved in multiple

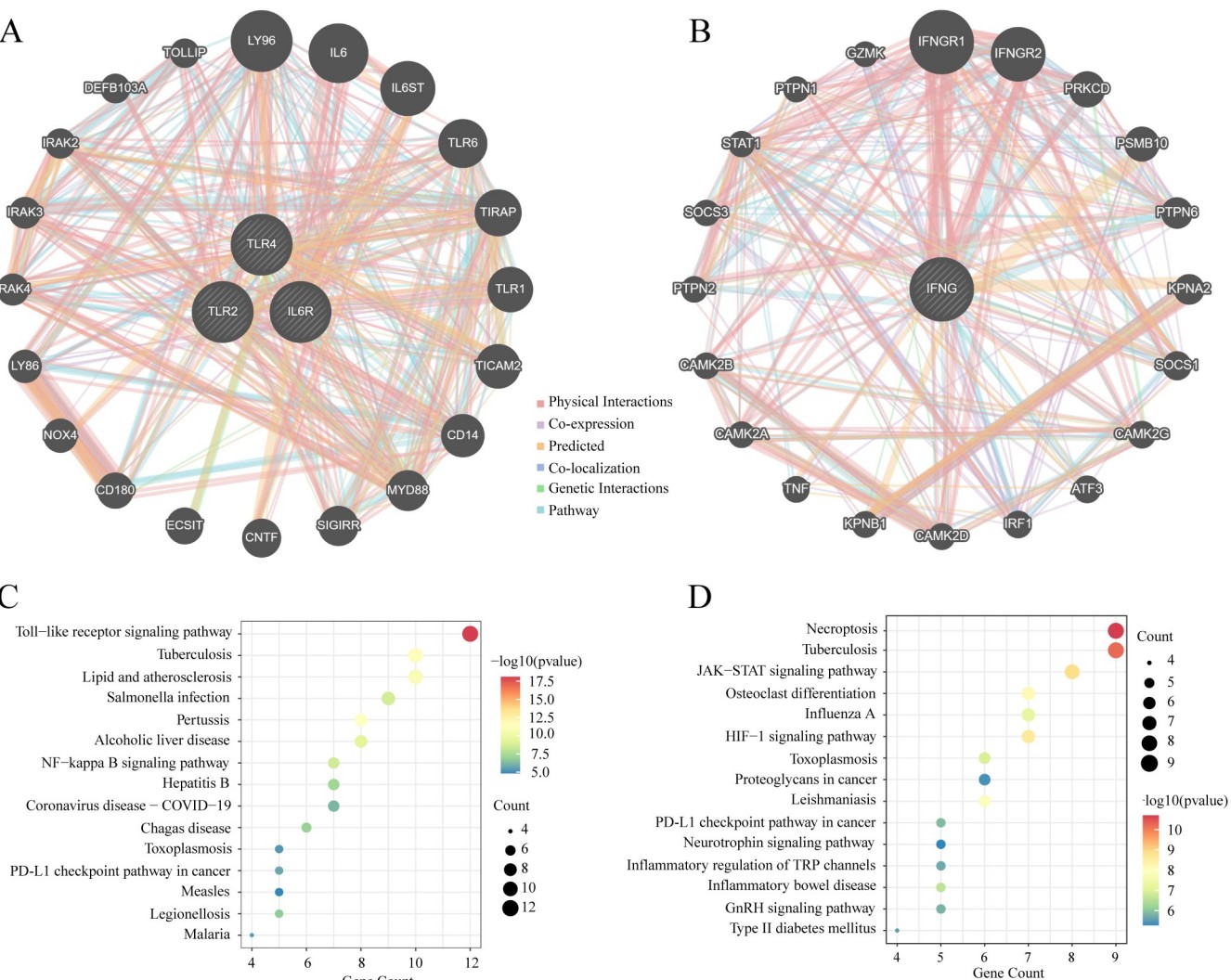

**Fig 6. The PPI network and KEGG analysis of the feature genes.** (A) The PPI network diagram of the upregulated genes. (B) The PPI network diagram of the downregulated genes. (C) The KEGG analysis of the upregulated and associated genes. (D) The KEGG analysis of the downregulated and associated genes.

inflammatory pathways and differed significantly from the signaling pathways enriched by the remaining 164,805 DEGs, suggesting that they might play a key role in CRS progression. Furthermore, the results showed that the YY1, RELA, EGR1, ETS2, and IRF1 transcription factors regulated these common genes. However, the role of these transcription factors in the development of the CRS associated with COVID-19 and as targets in disease treatment remains unclear. Therefore, further research on the transcription regulatory function of these 34 common genes may be beneficial for elucidating the CRS mechanism associated with COVID-19.

The IL-6 ligand of IL-6R is vital for cytokine storm mediation. Binding to the IL-6R ligand can cause various biological effects by activating the Janus kinase (JAK) signal, including naïve T cell maturation into effector T cells, inducing the secretion of various cytokines and chemokines by vascular endothelial cells, and activating coagulation cascade reactions [25, 26]. Some studies have shown that IL-6 maintains elevated levels in the peripheral blood of COVID-19 patients over long periods [27, 28], possibly serving as a predictive biomarker for disease severity [29]. A retrospective observational study showed that treatment with the biological agent

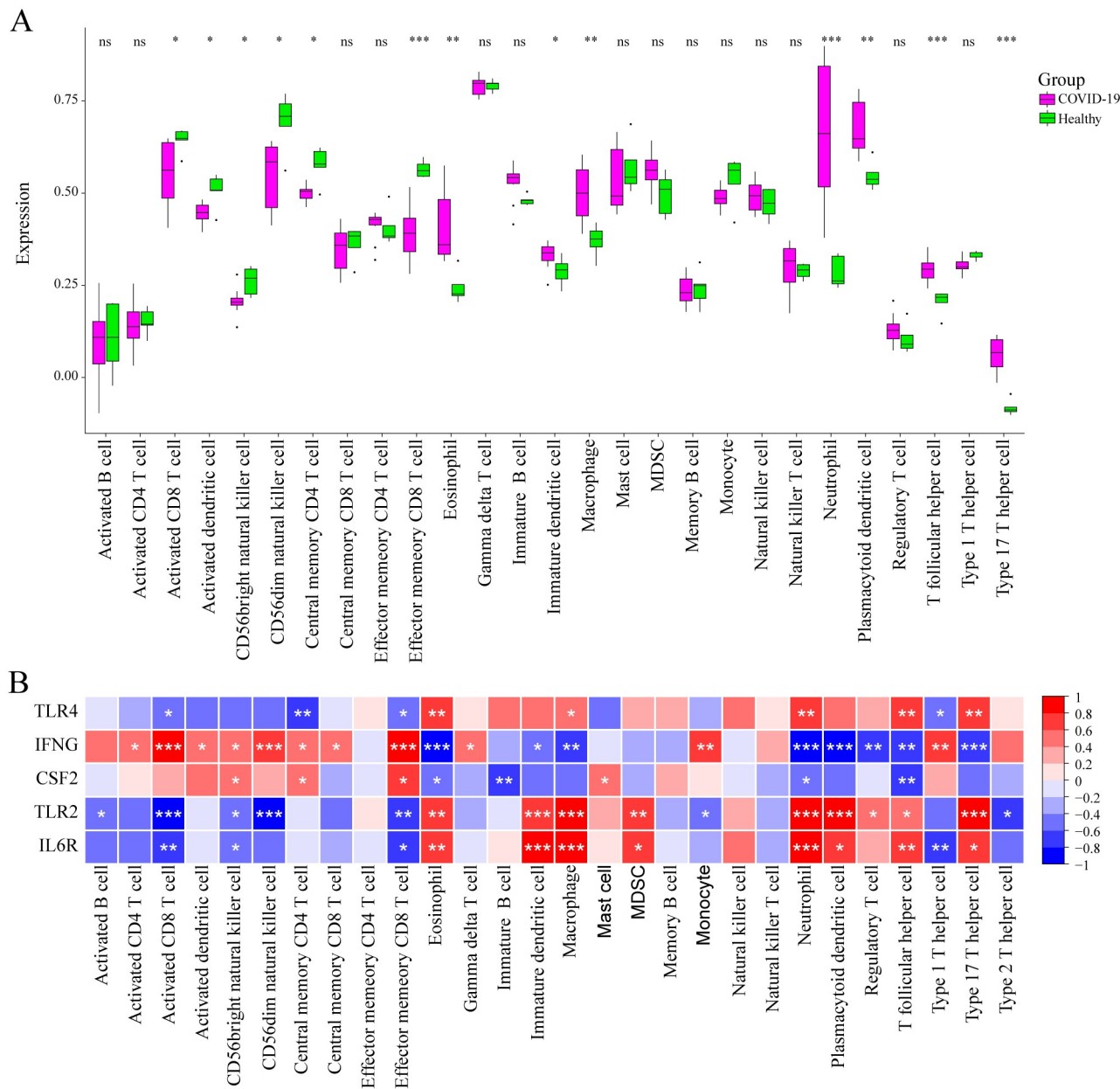

**Fig 7. Immune infiltration analysis.** (A) A comparison between the immune cell counts of the COVID-19 and healthy groups. * indicates p<0.05, ** indicates p<0.01, and *** indicates p<0.001. (B) The effect of higher feature gene expression on the immune cells. * indicates p<0.05, ** indicates p<0.01, and *** indicates p< 0.001.

Tocilizumab, which targets IL-6R, was effective in 46.7% of critically ill COVID-19 patients [30]. However, the role of IL-6R in the CRS associated with COVID-19 and as a treatment target remains unclear. In this study, bioinformatics integration showed that the IL-6R gene expression level was significantly higher in COVID-19 peripheral blood monocytes than in the healthy controls. Predictive analysis indicated that the Tocilizumab, Sarilumab, and Satalizumab biological agents effectively alleviated the biological effect of IL-6R. These results elucidate the role of IL-6R in the CRS pathogenesis associated with COVID-19 and as a treatment target.

**Table 1. Analysis of potential drugs targeting key genes.**

| DRUGBANK ID | NAME | DRUG GROUP | ACTIONS | GENE |
|---|---|---|---|---|
| DB06273 | Tocilizumab | approved | inhibitorantibody | IL6R |
| DB11767 | Sarilumab | approved, investigational | antagonistantibody | IL6R |
| DB15762 | Satralizumab | approved | binderantibody | IL6R |
| DB01296 | Glucosamine | approved, investigational | inhibitor | IFNG |
| DB05110 | VIR201 | investigational | - | IFNG |
| DB05111 | Fontolizumab | investigational | - | IFNG |
| DB01250 | Olsalazine | approved | - | IFNG |
| DB10770 | Foreskin fibroblast (neonatal) | approved | agonist | IFNG |
| DB10772 | Foreskinkeratinocyte (neonatal) | approved | agonist | IFNG |
| DB14724 | Emapalumab | approved, investigational | neutralizer | IFNG |
| DB04933 | Eritoran | investigational | - | TLR4 |
| DB06447 | E5531 | investigational | - | TLR4 |
| DB03017 | Lauric acid | approved, experimental | - | TLR4 |
| DB02767 | (R)-3-hydroxytetradecanoic acid | experimental | - | TLR4 |
| DB08231 | Myristic acid | experimental | - | TLR4 |
| DB01183 | Naloxone | approved, vet_approved | inhibitor | TLR4 |
| DB13615 | Mifamurtide | approved, experimental | ligand | TLR4 |
| DB11193 | Papain | approved | activator | TLR4 |
| DB00924 | Cyclobenzaprine | approved | inhibitor | TLR4 |
| DB00210 | Adapalene | approved | antagonist | TLR2 |
| DB05475 | Golotimod | investigational | - | TLR4/TLR2 |
| DB00045 | Lyme disease vaccine | approved, withdrawn | other/unknown | TLR2 |
| DB16474 | Pam2csk4 | investigational | agonist | TLR2 |
| DB03963 | S-(Dimethylarsenic)Cysteine | experimental | - | TLR2 |
| DB11601 | Tuberculin purified protein derivative | approved | ligand | TLR2 |

Toll-like receptors (TLRs) are key regulators of the innate immune system. It helps to identify self- and non-self-molecules, ultimately eliminating the non-self. Numerous studies have suggested that endosomal TLRs, mainly TLR3, TLR7, TLR8, and TLR4, play a role in CRS induction. TLR7/8 recognizes SARS-COV-2, and when it replicates to dsRNA, it is recognized by TLR3, driving inflammatory signaling, such as NF-κB and MAPK [31]. However, the role of TLR4 in the CRS associated with COVID-19 remains unclear. The persistently high interleukin-6 (IL-6) and tumor necrosis factor-α (TNF-α) levels in severe COVID-19 patients are important products of toll-like receptor 4 (TLR4) signaling. Therefore, believe that this viral infection and the subsequent organ damage may stimulate the toll-like receptor 4 (TLR4) pathways, increasing inflammatory cytokine production [32, 33]. Moreover, studies have shown that the TLR4 expression levels are significantly higher in the peripheral blood neutrophils and monocytes of severe COVID-19 patients than in healthy controls [31, 34]. This study indicated that the TLR4 expression levels were higher in the peripheral blood monocytes of COVID-19 patients with associated CRS than in healthy controls and revealed potential therapeutic drugs for targeting TLR4. These results confirm the role of TLR4 in the CRS pathogenesis associated with COVID-19 and its potential value as a treatment target.

Toll-like receptors 2 (TLR2), such as TLR4, belong to the TLRs protein family and can recognize β-defensins, heat shock and surfactant proteins, and high mobility group box 1 (HMGB1) proteins [35]. Studies have shown that the expression of TLR2 and its downstream MYD88 is correlated with COVID-19 severity and can sense the SARS-CoV-2 envelope protein to produce inflammatory cytokines [36]. Moreover, animal and in vitro cell experiments

showed that the SARS-CoV-2 spike protein caused inflammation via TLR2-dependent NF-κB pathway activation [37, 38]. One of its major ligands, HMGB1, is significantly elevated in the serum of COVID-19 patients and is associated with disease severity and CRS development [39]. However, the TLR2 action mechanism in the CRS associated with COVID-19 requires further confirmation. The genomic analysis of different data sets showed significantly elevated TLR2 expression in the peripheral blood mononuclear cells of COVID-19 patients, while potential therapeutic drug analysis was performed for this target. These results help clarify the mechanism behind TLR2 occurrence in the CRS associated with COVID-19.

Interferon-gamma (IFNG) is produced by natural killer cells, macrophages, and innate immunity effector cells, as well as the Th1 and CD8[+]T cells that participate in the adaptative response [40]. IFNG displays antiviral, antitumor, and immunomodulatory properties. The role of IFNG in COVID-19 progression remains unclear. Although reports have shown that the serum IFNG level in patients with CRS associated with COVID-19 is significantly lower than that in healthy controls [41], the therapeutic effect of IFNG injection on patients with severe COVID-19 remains controversial [40]. This study indicated that IFNG gene expression in the peripheral blood mononuclear cells of patients with CRS associated with COVID-19 was significantly lower than in healthy controls. The drugs associated with this target were analyzed. These results help elucidate the role of IFNG in the development of the CRS associated with COVID-19.

In conclusion, this study revealed that IL-6R, TLR4, TLR2, and IFNG may be potential pathogenic genes in the CRS associated with COVID-19. These findings help clarify the CRS pathogenesis associated with COVID-19. However, given the data collection limitations in the database and the need for continuous improvement of the methods used for big data analysis, more large sample prospective studies are required to confirm these results.

## Conclusion

IL6R, TLR4, TLR2, and IFNG may be potential pathogenic genes and therapeutic targets for the CRS associated with COVID-19.

## Supporting information

**S1 Method.**
(DOC)

**S1 Fig.**
(TIF)

**S1 Table. Comparison of enrichment analysis between 34 DEGs and remaining 6916 DEGs.**
(DOC)

## Acknowledgments

Thanks to all the colleagues who contributed to this research.

## Author Contributions

**Conceptualization:** Bing Yang, Meijun Pan.

**Data curation:** Kai Feng, Xue Wu.

**Methodology:** Kai Feng, Xue Wu.

**Project administration:** Fang Yang, Peng Yang.

**Software:** Bing Yang, Meijun Pan.

**Supervision:** Kai Feng, Xue Wu.

**Validation:** Bing Yang, Meijun Pan.

**Visualization:** Bing Yang, Meijun Pan.

**Writing – original draft:** Bing Yang, Meijun Pan.

**Writing – review & editing:** Fang Yang, Peng Yang.

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
