## [Decision Letter · Decision Letter 0]

3 Oct 2023

PONE-D-23-22764Identification of the Feature Genes involved in Cytokine Release Syndrome in COVID-19PLOS ONE

Dear Dr. Yang,

Thank you for submitting your manuscript to PLOS ONE. After careful consideration, we feel that it has merit but does not fully meet PLOS ONE’s publication criteria as it currently stands. Therefore, we invite you to submit a revised version of the manuscript that addresses the points raised during the review process.

We look forward to receiving your revised manuscript.

Kind regards,

Salman Sadullah Usmani, Ph.D.

Academic Editor

PLOS ONE

Journal Requirements:

Additional Editor Comments:

Dear Authors,

Your manuscript has been seen by 3 reviwers, and two of them raised substanital concern about the dataset used in the study, and I too agree with them. Given the importance of this study, I would like to see the effect on larger dataset. We find the study subject interested and worthwhile. Therefore, If you add the more dimesnion to the study, and answer the queries raised by all three reviwers, the study will be more suitable having indepth effect.

Reviewers' comments:

Reviewer's Responses to Questions

**Comments to the Author**

1. Is the manuscript technically sound, and do the data support the conclusions?

Reviewer #1: Yes

Reviewer #2: Partly

Reviewer #3: Partly

2. Has the statistical analysis been performed appropriately and rigorously? 

Reviewer #1: Yes

Reviewer #2: Yes

Reviewer #3: Yes

3. Have the authors made all data underlying the findings in their manuscript fully available?

Reviewer #1: Yes

Reviewer #2: Yes

Reviewer #3: Yes

4. Is the manuscript presented in an intelligible fashion and written in standard English?

Reviewer #1: Yes

Reviewer #2: Yes

Reviewer #3: Yes

5. Review Comments to the Author

Reviewer #1: The manuscript " Identification of the Feature Genes involved in Cytokine Release Syndrome in COVID-19” offers valuable insights into potential pathogenic genes associated with CRS in COVID-19 and suggests therapeutic targets.The manuscript addresses an important topic related to COVID-19, specifically the identification of feature genes associated with Cytokine Release Syndrome (CRS). The study employs various bioinformatics techniques and datasets to identify potential pathogenic genes and therapeutic targets. Overall, the manuscript is well-structured, and the research objectives are clear. The manuscript appropriately acknowledges the limitations, including data collection limitations and the need for further experiments. Expanding on these limitations and suggesting potential directions for future research would enhance the manuscript's completeness. Addressing the specific minor comments mentioned would improve the manuscript's overall quality and impact.

Minor comments:

1. While the manuscript is generally clear, there are some grammatical issues and awkward sentence structures. A thorough proofreading for language and clarity is recommended especially to look for misspellings like “interlaukin-6 production” line 171.

2. For “Common gene acquisition and functional enrichment” section line 161 onwards. It It would be better to define the meaning of “these common genes were involved in multiple biological processes during COVID-19 progression”. Since we are selecting only 34 common genes, is the Figure 2A only result of these gene or more than these differentially expressed (DE) genes. If only from this small group of genes, we should also look on other pathways from rest of DE genes and compare with these pathways from these 34 gene lists.

3. Figures can be improved in quality as well as proper labelling. The figure legends needs to be explained in more detail so that they are clear and do not need to look into the text for their explanation.

4. The code or scripts used in this work should be made available for scientific community for reproducibility of the results.

5. In the discussion section, more emphasis should be given to the found results rather than repeating the results found in Results section. How these are connected to the conditions related to the COVID and how can explore them!

Reviewer #2: In this study, the authors attempted to find the feature genes involved in cytokine release syndrome (CRS). Their research highlights the significance of finding potential pathogenic genes associated with CRS in COVID-19.The general concept and findings of this study are intriguing, and further research based on these findings

may contribute to a better understanding and control of COVID-19 progression into CRS. However, I have some queries about the study. I feel the authors need to carefully mention them in the manuscript.

Major

1) The sample size for COVID-19 gene expression is very small. Although the results are intriguing, I would recommend replicating them using an independent dataset.

2) How many of the 34 common genes between COVID-19 and CRS have already been linked to COVID-19 progression into CRS? I believe, authors should elaborate on this point in the discussion section. A balanced overview of the literature on COVID-19 progression into CRS is also missing.

Minor

1) Methods' section on feature gene acquisition needs to be rewritten.

2)The conclusion section should be revised. The last paragraph of the discussion section should be included in the conclusion section.

3) What do the letters CC and MF stand for? Authors must use their full name on the first occasion.

Reviewer #3: Comments to authors

The manuscript entitled “Identification of the Feature Genes involved in Cytokine Release Syndrome in COVID-19” presents a meticulous bioinformatics analysis aimed at identifying potential pathogenic genes associated with Cytokine Release Syndrome (CRS) in COVID-19 patients. The importance of this research cannot be overstated as it addresses a critical aspect of the pandemic, shedding light on the molecular mechanisms underlying severe inflammatory responses, which is crucial for the development of targeted therapeutic interventions.

The experimental design is well-structured and methodically executed, employing a range of bioinformatics tools and databases to analyze and validate the differentially expressed genes (DEGs) associated with CRS. The utilization of multiple datasets and validation through different algorithms like Betweenness and MCC, alongside enrichment analyses, adds a layer of robustness to the findings. Most of the results and analysis are presented in a logical sequence, culminating in the identification of core genes significantly associated with CRS in COVID-19 patients. This manuscript substantially contributes to the existing body of knowledge and has the potential to guide future research in this domain. However, one major and a few minor areas require attention before publication.

1. The data set taken could be more extensive. I strongly recommend that the authors to consider utilizing a larger dataset for analysis, given that the Gene Expression Omnibus (GEO) database likely contains data from a more extensive pool of individuals. Employing a larger dataset could provide more robust insights and enhance the generalizability of the findings. If leveraging a larger dataset is not feasible, I would appreciate a detailed explanation regarding the constraints or rationale behind selecting data from only ten individuals. Furthermore, it would be beneficial to understand if specific criteria guided this selection or if any consideration was given to pooling data from multiple datasets to achieve a larger sample size.

2. The authors have to provide more details in the material method section. For example, which PCA package and its version were used for PCA analysis?

3. Rewrite lines 66-69; “Cytokines are soluble polypeptide proteins secreted by immune cells, which play an intercellular regulatory role by binding corresponding receptors, controlling the inflammatory response, cell growth, differentiation and maturation, and immune response.”

Other cells can also produce cytokines, not just immune cells.

https://www.ncbi.nlm.nih.gov/pmc/articles/PMC2785020/

1) Some typo errors:

1. Line 69: response, cell growth, differentiation, and maturation, and immune response..

i) 2. In line 126, a full stop needs to be included.

ii) The format used for referring figures is not uniform. Fig. 1A, B, C are used in literature vs. a, b, c are used in the actual figure. The authors should correct it.

iii) Need to include references in lines 57-60.

iv) replace the term “including” with “namely” (please see line number 179)

v) In line 162, replace “CRS-related disease genes” with “CRS-related genes”.

vi) Re-write the sentence, “These results suggest that COVID-19 contains several genes that accelerate the progression to CRS.” The term contains is inappropriate in this context.

4. I suggest that authors reframe the sentence in lines 82-83, as there are few proposed hypotheses relating COVID-19 and CRS. The authors should also discuss similar hypotheses, supporting or contradicting the findings of this study.

https://www.ncbi.nlm.nih.gov/pmc/articles/PMC7919105/

https://www.ncbi.nlm.nih.gov/pmc/articles/PMC7151347/

5. The authors should rewrite the section “Feature genes acquisition”. In its present form, it looks more like a bullet-point format.

Here is a suggestion:

Common genes were imported into STRING 11.5 (https://cn.string-db.org/), and within the platform, the species was set to "Homo sapiens" (human) for the construction and visualization of a Protein-Protein Interaction (PPI) network. The results were then exported in TSV format and imported into Cytoscape 3.8.2 software, where a PPI network diagram was constructed. Following this, topology analysis was performed on the network, and core genes were screened using the Betweenness and Matthews Correlation Coefficient (MCC) algorithms.

6. The discussion section needs to be more detailed, and I suggest the authors completely rewrite the discussion section. Authors should discuss their findings along with other studies conducted on humans, mice or at least in vitro data.

For example, in line 256-257, “Various studies have shown that these signaling pathways also play a crucial role in the COVID-19 progression to CRS [27-29].” The authors should explain and discuss this statement in the context of already published literature.

6. PLOS authors have the option to publish the peer review history of their article (what does this mean?). If published, this will include your full peer review and any attached files.

Reviewer #1: **Yes: **Dr. Showkat Dar

Reviewer #2: No

Reviewer #3: **Yes: **Shoeb Ikhlas

---

## [Author Response · Author response to Decision Letter 0]

20 Nov 2023

Dear Editor and Reviewers:

We appreciate your consideration and insightful comments concerning our manuscript entitled "Identification of the Feature Genes involved in Cytokine Release Syndrome in COVID-19" (EMID:813efee4f8c9e179). The comments are valuable and helpful for improving the quality and readability of our paper and are important for guiding our future research. We have studied the comments carefully and revised the paper accordingly. Below are our point-by-point revisions according to the reviewers' suggestions.

Reviewer #1: 

The manuscript " Identification of the Feature Genes involved in Cytokine Release Syndrome in COVID-19” offers valuable insights into potential pathogenic genes associated with CRS in COVID-19 and suggests therapeutic targets.The manuscript addresses an important topic related to COVID-19, specifically the identification of feature genes associated with Cytokine Release Syndrome (CRS). The study employs various bioinformatics techniques and datasets to identify potential pathogenic genes and therapeutic targets. Overall, the manuscript is well-structured, and the research objectives are clear. The manuscript appropriately acknowledges the limitations, including data collection limitations and the need for further experiments. Expanding on these limitations and suggesting potential directions for future research would enhance the manuscript's completeness. Addressing the specific minor comments mentioned would improve the manuscript's overall quality and impact.

 Question 1: While the manuscript is generally clear, there are some grammatical issues and awkward sentence structures. A thorough proofreading for language and clarity is recommended especially to look for misspellings like "interlaukin-6 production" line 171.

 Response1: Thank you for your suggestion. We have hired native English-speaking professors to comprehensively revise the manuscript, including the text, materials, methods, and discussion. Misspelled errors like "Interleukin-6" have been corrected and are located in line 199 of the revised manuscript.

 Question 2: For “Common gene acquisition and functional enrichment” section line 161 onwards. It It would be better to define the meaning of “these common genes were involved in multiple biological processes during COVID-19 progression”. Since we are selecting only 34 common genes, is the Figure 2A only result of these gene or more than these differentially expressed (DE) genes. If only from this small group of genes, we should also look on other pathways from rest of DE genes and compare with these pathways from these 34 gene lists.

Response 2: Thank you for your suggestion. These suggestions help to better understand the role of common DEGs. Figure 2 shows 34 common DEGs and their enrichment analysis. In the revised manuscript, we added GO and KEGG enrichment analyses for the remaining 6916 DEGs and compared them with the 34 common DEG enrichment pathways via a list. The supplementary text is located in lines 200-203 and marked in red. Details are presented in S Fig 1 and S Table 1.

 Question 3: Figures can be improved in quality as well as proper labelling. The figure legends needs to be explained in more detail so that they are clear and do not need to look into the text for their explanation.

 Response 3: Thank you for your question. These suggestions are vital in helping improve the written quality of the figure legends. In the revised manuscript, we have redrawn the six figures in the original manuscript and made appropriate modifications to the figure markers and legends. The details are presented in lines 587-606, and the supplementary content is marked in red.

 Question4: The code or scripts used in this work should be made available for scientific community for reproducibility of the results.

 Response 4: Thank you for your question. In the revised manuscript, we have added all the code and scripts used for this study and placed them in the S1-8 method. Thank you again for your question.

Question 5：In the discussion section, more emphasis should be given to the found results rather than repeating the results found in Results section. How these are connected to the conditions related to the COVID and how can explore them!

Response 5: Thank you for your constructive feedback. This is crucial for improving the quality of our manuscript and the written level of our paper. We have rewritten the discussion section in the revised manuscript and cited new references. Thank you again for your help and hard work on the manuscript.

Reviewer #2: 

In this study, the authors attempted to find the feature genes involved in cytokine release syndrome (CRS). Their research highlights the significance of finding potential pathogenic genes associated with CRS in COVID-19.The general concept and findings of this study are intriguing, and further research based on these findings

may contribute to a better understanding and control of COVID-19 progression into CRS. However, I have some queries about the study. I feel the authors need to carefully mention them in the manuscript.

Major

 Question1): The sample size for COVID-19 gene expression is very small. Although the results are intriguing, I would recommend replicating them using an independent dataset.

Response1): Thank you for your question. The sample size of the dataset used in this study is indeed relatively small. However, the sample size of the COVID-19 peripheral blood monocyte genomics dataset collected in the GEO database is very small. We have attempted to integrate these individual datasets to increase sample data. However, due to the significant heterogeneity of different datasets, integrating datasets is very difficult, and the results obtained from the integrated data analysis may not necessarily reflect the true situation of the disease. To obtain relatively reliable results in this study, we selected the GSE164805 and GSE171110 datasets, whose PCA analysis results showed significant differences between the disease and healthy groups, as the research subjects. GSE164805 was used as the training set to screen common DEGs related to CRS, and GSE171110 was employed as the validation set to verify whether the screened core genes displayed similar expression changes to the training set. In theory, the common results obtained by analyzing the training and validation sets should explain the problem to a certain extent. Of course, the reliability of the research results still needs to be supported by prospective, large-scale clinical studies and animal experiments.

Question 2): How many of the 34 common genes between COVID-19 and CRS have already been linked to COVID-19 progression into CRS? I believe, authors should elaborate on this point in the discussion section. A balanced overview of the literature on COVID-19 progression into CRS is also missing.

 Response2): Thank you for your suggestion. These suggestions are critical for improving the quality of our manuscript and also help us improve our analytical skills and language expression skills. In the revised manuscript, we elaborated in detail the relationship between 34 common genes and COVID-19-related CRS progression and supplemented the literature on CRS-related progression. The details are presented in lines 301-312 and marked in red.

Minor

 Question1)： Methods' section on feature gene acquisition needs to be rewritten.

Response1): Thank you for your question. The method for feature genes acquisition has been rewritten in the revised manuscript. The details are located in lines 154-159 and marked in red.

Question2)：The conclusion section should be revised. The last paragraph of the discussion section should be included in the conclusion section.

Response2): Thank you for your question. The conclusion section has been amended in the revised manuscript and is presented in the last paragraph of the discussion. The details are located in lines 49-50 and 371-376 and marked in red.

Question3)：What do the letters CC and MF stand for? Authors must use their full name on the first occasion.

Response3): Thank you for your question. In the revised manuscript, we have added the full names of CC and MF. The details are in lines 197-198 and are marked in red.

Reviewer #3: 

The manuscript entitled “Identification of the Feature Genes involved in Cytokine Release Syndrome in COVID-19” presents a meticulous bioinformatics analysis aimed at identifying potential pathogenic genes associated with Cytokine Release Syndrome (CRS) in COVID-19 patients. The importance of this research cannot be overstated as it addresses a critical aspect of the pandemic, shedding light on the molecular mechanisms underlying severe inflammatory responses, which is crucial for the development of targeted therapeutic interventions.

Question1: The data set taken could be more extensive. I strongly recommend that the authors to consider utilizing a larger dataset for analysis, given that the Gene Expression Omnibus (GEO) database likely contains data from a more extensive pool of individuals. Employing a larger dataset could provide more robust insights and enhance the generalizability of the findings. If leveraging a larger dataset is not feasible, I would appreciate a detailed explanation regarding the constraints or rationale behind selecting data from only ten individuals. Furthermore, it would be beneficial to understand if specific criteria guided this selection or if any consideration was given to pooling data from multiple datasets to achieve a larger sample size.

Response 1: Thank you for your question. The sample size of the dataset used in this study is indeed relatively small. However, the sample size of the COVID-19 peripheral blood monocyte genomics dataset collected in the GEO database is very small. We have attempted to integrate these individual datasets to obtain more comprehensive sample data. However, due to the significant heterogeneity of different datasets, integrating datasets is very difficult, and the results obtained from the integrated data analysis may not necessarily reflect the true situation of the disease. To obtain relatively reliable results in this study, we selected the GSE164805 and GSE171110 datasets, whose PCA analysis results showed significant differences between the disease and healthy groups, as the research subjects. GSE164805 was used as the training set to screen common DEGs related to CRS, and GSE171110 was employed as the validation set to verify whether the screened core genes displayed similar expression changes to the training set. In theory, the common results obtained by analyzing the training and the validation sets should explain the problem to a certain extent. Of course, the reliability of the research results still needs to be supported by prospective, large-scale clinical studies and animal experiments.

 Question2:. The authors have to provide more details in the material method section. For example, which PCA package and its version were used for PCA analysis?

Response2: Thank you for your question. More details have been added to the materials and methods of the revised manuscript and are marked in red(127 line). The code or script used in this work has been placed in the supplemental method.

 Question3. Rewrite lines 66-69; “Cytokines are soluble polypeptide proteins secreted by immune cells, which play an intercellular regulatory role by binding corresponding receptors, controlling the inflammatory response, cell growth, differentiation and maturation, and immune response.”Other cells can also produce cytokines, not just immune cells.

Response3: Thank you for pointing out this issue to us. In the original manuscript, the cytokine definition was described inaccurately. In the revised manuscript, we changed "immune cells" to "immune cells and non-immune cells, such as endothelial cells, epidermal cells, and fibroblasts." The additional text is presented in lines 83-86 and marked in red.

Some typo errors:

Question 1:Line 69: response, cell growth, differentiation, and maturation, and immune response.

Response 1: Thank you for your question. The error has been corrected in the revised manuscript. The details are located in line 85.

Question i): 2. In line 126, a full stop needs to be included.

Response i): Thank you for your question. The error has been corrected in the revised manuscript. The details are located in line 152.

Question ii): The format used for referring figures is not uniform. Fig. 1A, B, C are used in literature vs. a, b, c are used in the actual figure. The authors should correct it.

Response ii): Thank you for your question. The error has been corrected in the revised manuscript. Please refer to Fig 1. 

Question iii) :Need to include references in lines 57-60.

Response iii): Thank you for your question. Two references have been added to the revised manuscript. The details are presented in lines 76 and 433-439 (references) and marked in red.

Question iv): replace the term “including” with “namely” (please see line number 179)

Response iv): Thank you for your question. In the revised manuscript, the term "including" has been replaced with "named." The details are located in line 210 and marked in red.

Question v):In line 162, replace “CRS-related disease genes” with “CRS-related genes”.

Response v): Thank you for your question. In the revised manuscript, the term "CRS-related disease genes" has been replaced with "CRS-related genes." The details are located in line 190 and marked in red.

Question vi) Re-write the sentence, “These results suggest that COVID-19 contains several genes that accelerate the progression to CRS.” The term contains is inappropriate in this context.

Response vi): Thank you for your question. We have rewritten the discussion section in the revised manuscript. Thank you again for your help and hard work on the manuscript.

Question 4: I suggest that authors reframe the sentence in lines 82-83, as there are few proposed hypotheses relating COVID-19 and CRS. The authors should also discuss similar hypotheses, supporting or contradicting the findings of this study.

 Response4: Thank you for your question. The sentence has been revised in the revised manuscript(lines 99-100), and the "discussion" around this hypothesis has been rewritten. 

 Question 5:The authors should rewrite the section “Feature genes acquisition”. In its present form, it looks more like a bullet-point format.

 Response 5: Thank you for your suggestion, which is beneficial for improving the quality of the manuscript and our language expression level. We have fully adopted your suggestions in the revised manuscript. The details are located in lines 153-158 and marked in red. Thank you again for your help and hard work on the manuscript.

 Question 6: The discussion section needs to be more detailed, and I suggest the authors completely rewrite the discussion section. Authors should discuss their findings along with other studies conducted on humans, mice or at least in vitro data.

Response 6: Thank you for your constructive feedback. This is crucial for improving the quality of our manuscript and the written level of our paper. We have rewritten the discussion section in the revised manuscript and cited new references. Thank you again for your help and hard work on the manuscript.

We greatly appreciate your efforts and the valuable input provided by the reviewers. The comments offered by the reviewers have proven to be quite helpful to us. They highlight the deficiencies of our manuscript, as well as the aspects requiring further elaboration. These comments are important in guiding our future research and help us with further improvements.

We have tried our best to improve the manuscript and extensively modified the original manuscript in response to the comments. These changes do not influence the content and framework of the paper. Although not all the changes are listed here, they are marked in red in the revised manuscript.

We thank you and the reviewers again for your help and hope the corrections meet with your approval.

Yours Sincerely,

Peng Yang

---

## [Decision Letter · Decision Letter 1]

5 Dec 2023

Identification of the Feature Genes involved in Cytokine Release Syndrome in COVID-19

PONE-D-23-22764R1

Dear Dr. Yang,

We’re pleased to inform you that your manuscript has been judged scientifically suitable for publication and will be formally accepted for publication once it meets all outstanding technical requirements.

Kind regards,

Salman Sadullah Usmani, Ph.D.

Academic Editor

PLOS ONE

Additional Editor Comments (optional):

Please check for any typos and minor suggestions from reviewer 2. If you believe that including these suggestions would benefit the manuscript, feel free to incorporate them at this stage. Alternatively, proofreading in its current form is also acceptable, as we trust that you will rectify any typos during the proofread.

Reviewers' comments:

Reviewer's Responses to Questions

**Comments to the Author**

1. If the authors have adequately addressed your comments raised in a previous round of review and you feel that this manuscript is now acceptable for publication, you may indicate that here to bypass the “Comments to the Author” section, enter your conflict of interest statement in the “Confidential to Editor” section, and submit your "Accept" recommendation.

Reviewer #1: All comments have been addressed

Reviewer #2: All comments have been addressed

Reviewer #3: All comments have been addressed

2. Is the manuscript technically sound, and do the data support the conclusions?

Reviewer #1: Yes

Reviewer #2: Yes

Reviewer #3: Yes

3. Has the statistical analysis been performed appropriately and rigorously? 

Reviewer #1: Yes

Reviewer #2: Yes

Reviewer #3: Yes

4. Have the authors made all data underlying the findings in their manuscript fully available?

Reviewer #1: Yes

Reviewer #2: Yes

Reviewer #3: Yes

5. Is the manuscript presented in an intelligible fashion and written in standard English?

Reviewer #1: Yes

Reviewer #2: Yes

Reviewer #3: Yes

6. Review Comments to the Author

Reviewer #1: (No Response)

Reviewer #2: 1) Please consistently use the abbreviated form of Cytokine release syndrome (CRS) after the first use. I can still see that this has not been changed.

2) I believe, there is no need to mention the code used is listed in S1 and S2 methods in each section. You can mention something like “the packaged and in-house codes used in this study are mentioned in supplementary material” at the end of methods section.

3) Please remove “GSE164805” from line 161 since it was used as a training set, not for validation.

4) Lines 199-200 need to be repharsed. What does positive regulation means? Use something like “controlled the regulation of cycline production, defense response and interleukin-6 production in the BP module”.

5) The last paragraph of discussion section should be part of conclusion section.

Reviewer #3: The revised manuscript effectively addresses the previously highlighted concerns and incorporates suggested improvements. The clarity of the experimental design and the meticulous execution of bioinformatics analyses remain commendable. The authors have successfully enhanced the manuscript's robustness by responding to comments, refining methodologies, and ensuring logical presentation of results. The comprehensive identification of core genes associated with Cytokine Release Syndrome in COVID-19 patients significantly contributes to the field. In my opinion, the revised manuscript is now ready for publication, offering valuable insights.

7. PLOS authors have the option to publish the peer review history of their article (what does this mean?). If published, this will include your full peer review and any attached files.

Reviewer #1: **Yes: **Showkat Dar

Reviewer #2: No

Reviewer #3: No

---

## [Editor Report · Acceptance letter]

18 Dec 2023

PONE-D-23-22764R1 

PLOS ONE

Dear Dr. Yang, 

I'm pleased to inform you that your manuscript has been deemed suitable for publication in PLOS ONE. Congratulations! Your manuscript is now being handed over to our production team.

Kind regards, 

on behalf of

Dr. Salman Sadullah Usmani 

Academic Editor

PLOS ONE